# Reverse Sap Flow from Fruit

**DOI:** 10.3390/plants15010105

**Published:** 2025-12-30

**Authors:** Yangfan Chai, Runqing Zhang, Qian Wang, Jiawei Pan, Yuanhao Wang, Yu Zou, Shuai Wang, Zhongyuan Hu, Xiangjiang Liu

**Affiliations:** 1College of Biosystems Engineering and Food Science, Zhejiang University, Hangzhou 310058, China; chaiyf@zju.edu.cn (Y.C.); wshuaizju@163.com (S.W.); 2College of Agriculture and Biotechnology, Zhejiang University, Hangzhou 310058, China; 22116201@zju.edu.cn (R.Z.); huzhongyuan@zju.edu.cn (Z.H.); 3College of Mechanical and Electrical Engineering, Xinjiang Agricultural University, Urumqi 830052, China; hebau_wangqian@163.com (Q.W.); pjw0610@163.com (J.P.); education_wyh@163.com (Y.W.); zouy9911@163.com (Y.Z.); 4State Key Laboratory of Agricultural Equipment Technology, Hangzhou 310058, China

**Keywords:** reverse sap flow, plant-wearable sensor, soil-plant-atmosphere continuum, water balance, source–sink theory

## Abstract

Sap flow serves as the primary carrier for water, nutrients, and signaling molecules, playing a crucial role in fruit development by delivering these essential constituents to the fruit. While the efflux of sap from fruit to other organs (termed reverse sap flow) has been observed in plants, its underlying mechanisms remain unclear due to a lack of effective methodologies for comprehensive studies. Here, we pioneered the integration of real-time sap flow measurements from novel plant-wearable sensors with synchronized environmental monitoring, establishing a multimodal data framework to systematically decode the endogenous causes and exogenous triggers of reverse sap flow in watermelon plants. Our experimental results reveal that plant water supply–consumption imbalance is the core endogenous cause of reverse sap flow, which is induced by two external triggers in the natural environment: rapid light intensity surges and soil drought. Furthermore, a long-term drought stress experiment illustrates that reverse sap flow from the fruit enhances the drought resistance of plants by adjusting water redistribution within the whole plant. This study challenges the unitary view of fruit solely as a “sink” in the traditional source–sink theory, further refines the understanding of the source–sink paradigm, and provides a novel mechanism and insight for plant drought tolerance strategies.

## 1. Introduction

Sap flow, the liquid movement within plant stems driven by water potential gradients, serves as the primary carrier for water, nutrients, and signaling molecules [1,2]. It constitutes a critical component of internal water transport within the soil-plant-atmosphere continuum (SPAC) [3], as shown in Figure 1A. This hydraulic system describes a pathway where liquid water is absorbed by roots from the soil, translocated via the stem’s sap flow to aerial tissues, and ultimately transpired through leaf stomata into the atmosphere. While sap flow in fruits is conventionally regarded as unidirectional, moving toward these terminal “sink” tissues to sustain growth [4,5], recent studies have documented instances of reverse sap flow (i.e., water efflux from fruits to vegetative organs) [6,7,8,9], challenging the long-standing paradigm of root-to-fruit water transport. For instance, Knoche et al. observed that xylem backflow from fruit to tree frequently occurred in the morning in developing European plums [6]. Keller et al. injected dye into grape berries and documented the movement of xylem-mobile dye from the berries back to the shoot during the ripening stage for sugar accumulation [7]. Similarly, Zhang et al. employed a comparable method and found that a portion of water flowed back from the maize cob and ear to the plant via the xylem during the grain dehydration phase [8]. Although these studies observed reverse sap flow across multiple crop species, they were unable to explain the underlying causes of this phenomenon due to the inability to nondestructively and accurately quantify the movement of sap flow in plants in real time. This knowledge gap is primarily due to the inherent nature of sap flow as a subtle, continuously changing process occurring deep within plant tissues, necessitating long-term, continuous, and highly accurate monitoring for systematic analysis of water dynamics, a capability currently lacking in effective measurement methods.

Existing sap flow measurement methods, while validated across diverse plant species, face intrinsic limitations that impede long-term, noninvasive field monitoring. Destructive methods, such as the stem excision technique [10,11], cause irreversible damage to plant stems. Conversely, nondestructive tracer-based methods, such as isotope labeling [12,13] and dye tracing [14,15], are unsuitable for long-term monitoring because the plant metabolizes the markers within a few days. Furthermore, nuclear magnetic resonance imaging (MRI) [16,17] lacks field applicability owing to its nonportability and operational complexity. Even classical heat-based sap flow meters [18,19] often fail to achieve long-term stable monitoring, particularly in small herbaceous plants, as their rigid probes mechanically mismatch living tissues and impose size restrictions, which compromise their stability and noninvasive deployment.

Recent advances in flexible electronics have unlocked new possibilities to address this challenge. Flexible electronics is a technology in which sensing components are fabricated on flexible, stretchable substrates, allowing electronic devices to maintain functionality under bending/stretching and to conformally adhere to irregular surfaces [20,21]. This capability effectively eliminates the physical damage inflicted on biological tissues by the mechanical mismatch inherent in rigid electronics [22,23]. Based on this technology, developed wearable biosensors can conformally integrate with bio-interfaces for high-fidelity physiological sensing. In the agricultural domain, plant-wearable sensors have already been used to track surface temperature [24,25], humidity [26,27], electrophysiological signals [28,29], growth rate [30,31], leaf water content [32,33], and volatile organic compounds (VOCs) [34,35] in plants. Leveraging this innovation, we previously developed the first plant-wearable sap flow sensor capable of in situ, nondestructively monitoring herbaceous plant sap flow. Deployment of these sensors on watermelon plants revealed a previously unknown day/night shift pattern of water allocation between the fruit and its adjacent branch [36]. Furthermore, during the watermelon’s full growth cycle, we also observed reverse sap flow in fruits, though its underlying causal mechanisms were unexplored [37].

Therefore, the objectives of this study were to elucidate the underlying causal mechanisms of reverse sap flow and to investigate its role in plant water physiology. Here, we pioneer the integration of real-time sap flow measurements from plant-wearable sensors with synchronized environmental monitoring, establishing a multimodal data framework to systematically decode the endogenous causes and exogenous triggers of reverse sap flow—a previously unaddressed knowledge gap. The experimental results demonstrate that reverse sap flow from the fruit occurs when the water consumption by plant shoots (Sap*_shoot_*) surpasses the water supply from the roots (Sap*_root_*) (Figure 1B), a deficit primarily induced by two environmental triggers in the field: rapid light intensity surges and soil drought. This establishes the supply–consumption difference, ΔSap = Sap*_root_* − Sap*_shoot_*, as a key indicator, where its sign dictates the direction of sap flow relative to the fruit. This study redefines fruits as not merely passive “sinks” but also transient “water reservoirs” that enhance environmental resilience through inter-organ water redistribution. Furthermore, this discovery integrates fruits into the whole-plant hydraulic network, broadening the understanding of plant drought response diversity.

## 2. Results

### 2.1. In Situ Monitoring of Water Distribution Within Watermelon Plants

As shown in Figure 2A, we deployed previously developed plant-wearable sap flow sensors on healthy watermelon plants (variety: 8424) three weeks after fruit set, for in situ monitoring of water transport within the plant. Our plant-wearable sap flow sensor comprises two aligned temperature sensors and a heater bonded to serpentine copper conductive tracks (Figure 2B), laminated onto a stretchable, air-permeable polydimethylsiloxane (PDMS) substrate (Appendix A). The sensing mechanism is shown in Figure 2C. Briefly, the heater induces a localized temperature increase, and in the presence of sap flow, most of the heat is carried downstream, resulting in a rapid temperature rise downstream and an anisotropic distribution along the stem. This temperature distribution is measured by the two symmetrically placed temperature sensors. The temperature difference (ΔT = T_downstream_ − T_upstream_) between the two sensors in response to varying sap flow rates enables precise determination of sap flow rate and direction, as depicted in Figure 2D. The ascending slope of the ΔT curve demonstrates a strong linear correlation with sap flow rate (R^2^ = 0.97). The detailed calibration of the sap flow sensor is shown in Appendix A.

Three sensors were strategically installed at key hydraulic positions of the watermelon plant (Figure 2A): basal stem—measuring root-derived water supply (Sap*_root_*), leaf branch—quantifying shoot transpiration-driven water consumption (Sap*_sh__oot_*), fruit branch—tracking bidirectional sap flow of fruits (Sap*_fruit_*). Figure 2E presents fluctuations in sap flow rates over 7 days in the watermelon plant, with normal plant growth observed during the monitoring period. Notably, our observations revealed diurnal reverse sap flow dynamics—where Sap*_fruit_* > 0 (blue curve) indicates water inflow to fruits and Sap*_fruit_* < 0 (orange curve) denotes reverse sap flow—occurring with near-daily regularity during daylight periods.

For mechanistic analysis, we analyzed the sap flow data collected during the daytime on a representative day (Day 4 in Figure 2E). Figure 2F, G illustrate the sap flow rates at three plant positions during the diurnal light period. During the morning (07:30–10:30), rapidly increasing light intensity and temperature triggered stomatal opening, resulting in a significant enhancement of transpiration that exceeded root water uptake capacity (Sap*_shoot_* > Sap*_root_*), as shown in Figure 2F. In this period, reverse sap flow (Sap*_fruit_* < 0) was often observed in the fruit (Figure 2G, orange bars). By midday (11:00–13:00), excessive light intensity and temperature induced stomatal closure as a protective response, transiently reducing shoot transpiration (Sap*_shoot_* < Sap*_root_*) and restoring fruit inflow (Sap*_fruit_* > 0). After that, declining light intensity and temperature reopened stomata initially, reigniting fruit reverse flow (13:00–16:00) until gradual stomatal closure in response to lower light intensity and temperature reestablished inflow to fruits (after 16:00). The difference between water supply and water consumption (ΔSap = Sap*_root_* − Sap*_shoot_*, blue and orange bars) was calculated in Figure 2G. We found that when ΔV turned negative (ΔSap < 0, indicating water consumption exceeded supply), the fruit sap flow rate (red curve) simultaneously became negative (Sap*_fruit_* < 0) in most cases. A high linear correlation (R^2^ = 0.87) between ΔSap and Sap*_fruit_* (Figure 2H) further confirmed that reverse sap flow systematically occurs when whole-plant water consumption surpasses root supply capacity, demonstrating that water supply–consumption imbalance governs fruit sap flow directionality.

### 2.2. Environmental Factors Impacting the Water Supply–Consumption Balance of Plants

The previous experiments have demonstrated that reverse sap flow occurs when water supply is less than water consumption (Sap*_root_* < Sap*_shoot_*), mobilizing fruit-stored water to compensate for shoot transpiration deficits through regulated outflow from fruits. The condition of Sap*_root_* < Sap*_shoot_* arises in two scenarios: (1) water consumption increases to exceed supply, or (2) water supply decreases to fall below consumption. To identify the environmental triggers underlying these dynamics, we simultaneously monitored sap flow rates at three hydraulic positions (basal stem, leaf branch, fruit branch) alongside key environmental parameters (light intensity, air temperature/humidity, soil water content). Our experiments revealed two primary environmental triggers that induce water supply–consumption imbalance and reverse sap flow (Figure 3A): (i) Rapid light intensity increase caused immediate stomatal opening, leading to a sharp rise in Sap*_shoot_* while Sap*_root_* remained relatively unchanged. As a result, Sap*_shoot_* exceeded Sap*_root_*, and Sap*_fruit_* shifted to a negative value (reverse flow). (ii) Soil water deficit prevented the roots from absorbing sufficient water, causing Sap*_root_* to drop significantly below Sap*_shoot_*, and Sap*_fruit_* became negative (reverse flow) as well.

Figure 3B illustrates the diurnal fruit sap flow rates over two days under adequate soil moisture conditions in response to solar radiation intensity. Specifically, when light intensity fluctuated little during the daylight period (06:00–18:00), the root water supply adequately met the transpiration demand driven by gradual stomatal adjustments. Consequently, virtually no reverse sap flow was observed (Day 1). In contrast, once light intensity increased significantly in a short time (Day 2, gray shaded areas), Sap*_shoot_* would rise suddenly and exceed Sap*_root_* (Sap*_shoot_* > Sap*_root_*). This imbalance caused the fruit sap flow rate to switch from positive to negative values, indicating the occurrence of reverse sap flow. During periods of low or absent light (before 6:00 and after 18:00), the reduced transpirational demand maintained water consumption below supply capacity (Sap*_shoot_* < Sap*_root_*), consistently yielding positive fruit sap flow rates (Sap*_fruit_* > 0). Figure 3C shows the relationship between the probability of reverse sap flow and the rate of light intensity increase (k). When k is small (<20 μmol m^−2^ s^−1^/min), the stomatal status remains stable without abrupt changes, allowing water supply to meet transpiration demand without requiring reverse sap flow from fruits. As k gradually increases, the enhanced rate of stomatal opening increases transpiration, leading to a higher likelihood of reverse sap flow occurring when the water supply is insufficient to meet the demand. As k reaches a critical threshold (= 80 μmol m^−2^ s^−1^/min), the subsequent rapid stomatal opening triggers sudden transpiration surges that markedly surpass root water supply capacity, resulting in a high probability (over 80%) of reverse sap flow from fruits. These results indicate that a greater rate of light intensity increase (k) critically elevates the probability of reverse sap flow occurrence.

Besides light intensity, soil water deficit also triggers reverse sap flow. As shown in Figure 3D, under adequate soil moisture conditions, intermittent reverse sap flow (Sap*_fruit_* < 0) occurred during certain daylight periods (gray shaded areas) due to Sap*_shoot_* > Sap*_root_* caused by drastic light intensity fluctuations, with Sap*_fruit_* > 0 at all other times. However, when soil moisture was extremely low, water supply was insufficient to meet consumption (Sap*_root_* < Sap*_shoot_*), causing reverse sap flow to persist throughout the entire high-sunlight period (7:30–15:30) to collectively sustain plant water consumption. During other periods (before 7:30 and after 15:30) with weak light intensity or no light, water consumption decreased significantly below the supply (Sap*_shoot_* < Sap*_root_*), and Sap*_fruit_* reverted to positive values (Sap*_fruit_* > 0). Figure 3E demonstrates the relationship between the daily duration of reverse sap flow (T) and soil water content: under ample soil moisture conditions (>80%), plant roots could absorb sufficient water from the soil, with intermittent daytime reverse sap flow occurring primarily due to drastic light intensity fluctuations, while T remained below 6 h; as soil water content progressively decreased, inadequate root water uptake failed to meet transpirational demand, leading to sustained reverse sap flow that caused a marked increase in T, which exceeded 16 h under severe drought conditions (soil water content < 20%). In conclusion, these results reveal that both light intensity and soil moisture are critical environmental factors influencing the water supply–consumption balance, thereby triggering reverse sap flow.

### 2.3. Fruit-Mediated Water Redistribution Enhances Plant Drought Resilience

After confirming that plants can respond to environmental stress via reverse sap flow from the fruit, we sought to further investigate whether fruits can act as temporary “water reservoirs” to enhance drought resistance. To address this, we conducted a comparative experiment using two healthy watermelon plants with similar growth conditions—one bearing fruit and one defruited—both subjected to prolonged drought stress.

As shown in Figure 4A, sap flow sensors were deployed at three positions on the fruit-bearing plant to monitor water dynamics while plant growth status was recorded daily. During the initial stage with sufficient soil moisture (Day 1), water primarily flowed from roots to fruits and shoots throughout most of the day (gray-shaded areas), with intermittent reverse sap flow from fruits occurring during certain daylight periods in response to Sap*_shoot_* > Sap*_root_* triggered by light intensity surge. As soil moisture gradually declined, water supply decreased markedly below consumption (Sap*_root_* < Sap*_shoot_*), resulting in sustained reverse sap flow from fruits to collectively support shoot transpiration during the high-light period (07:30–15:30, Day 6). Under severe drought (Day 8), roots failed to absorb sufficient water, leading to a notable phenomenon: reverse sap flow from fruits no longer supplied shoots (Sap*_shoot_* = 0) but instead directed toward the roots (Sap*_root_* < 0) to maintain the plant alive and support the leaves near the roots during the high-light period (gray-shaded areas). Concurrently, vines between fruits and apical shoots withered, while sections from roots to fruits remained viable. By day 12, both Sap*_fruit_* and Sap*_root_* approached zero, and the entire fruit-bearing plant ultimately withered.

In the defruited plant (Figure 4B), sap flow sensors were placed on two stem positions, excluding the fruit branch. Sap*_root_* and Sap*_shoot_* gradually decreased in synchrony with soil drought progression (Day 1 to Day 6), maintaining Sap*_root_* > Sap*_shoot_* throughout. As drought severity increased, plant roots became incapable of extracting soil water, driving both hydraulic parameters (Sap*_root_* and Sap*_shoot_*) toward zero and resulting in complete plant wilting by Day 8.

Overall, the fruit-bearing plant survived four days longer than its defruited counterpart under prolonged drought stress. These findings suggest that reverse sap flow from fruits, triggered by water supply–consumption imbalances, can buffer water deficits and sustain physiological function under drought stress, serving as a critical drought adaptation strategy.

## 3. Discussion

Reverse sap flow from the fruit, a hydraulic phenomenon challenging conventional unidirectional water transport paradigms of fruits, has been observed in recent studies. However, the underlying mechanisms driving this hydraulic reversal remain poorly understood, largely due to limitations in monitoring techniques. This study pioneers a multimodal approach, integrating real-time sap flow rates measured by novel plant-wearable sensors from multiple key plant positions with concurrent environmental parameters, to systematically decode the endogenous causes and exogenous triggers of reverse sap flow.

We deployed self-developed, plant-wearable sap flow sensors at three critical positions—basal stem, leaf branch, and fruit branch—of watermelon plants to monitor long-term internal water dynamics, observing daytime reverse flow events (Figure 2E). Analysis of a representative day demonstrated that the water supply–consumption imbalance determines fruit sap flow direction: when root water uptake exceeds shoot transpiration rates (Sap*_root_* > Sap*_shoot_*), surplus water is allocated to fruits (inflow); conversely, when Sap*_root_* < Sap*_shoot_*, water is withdrawn from fruits (outflow) to compensate for shoot transpiration deficits (Figure 1 and Figure 2G). The strong correlation observed between Sap_fruit_ and ΔSap (Sap*_root_* − Sap*_shoot_*) further supports this conclusion (Figure 2H).

Building on this foundation, we investigated environmental triggers of reverse sap flow in field-grown plants under natural conditions. Coordinated monitoring of environmental parameters and plant internal water dynamics identified two distinct mechanistic triggers:

(i) Light-triggered transient reverse sap flow: Under adequate soil moisture conditions, a rapid increase in light intensity induced immediate stomatal opening, abruptly elevating transpiration beyond root water supply (Sap*_shoot_* > Sap*_root_*), and resulting in intermittent reverse flow during the daytime (Figure 3B). Crucially, the probability of this event significantly increased with a higher rate of increase in light intensity (Figure 3C).

(ii) Drought-induced sustained reverse sap flow: Under soil water deficit, root water uptake was insufficient to meet daytime transpiration demands (Sap*_root_* < Sap*_shoot_*), triggering sustained reverse flow from fruits in the daylight period (Figure 3D). Furthermore, the duration of this sustained reverse flow was inversely proportional to soil moisture content, lasting longer as the soil humidity decreased (Figure 3E).

To evaluate the physiological significance of this phenomenon, comparative experiments with fruit-bearing vs. fruit-removed plants under drought stress demonstrated that fruits can temporarily function as “water reservoirs”, supplying water via reverse sap flow to support root function and thereby enhancing the plant’s drought resistance (Figure 4). The physiological basis for this observation lies in the fundamental principle of the water potential gradient, which dictates that water invariably moves from areas of higher to lower potential. Under severe drought, when the roots are unable to absorb water from the soil, their water potential plummets. Once the root water potential falls substantially below that of the fruit—which acts as a storage capacitor—the hydraulic gradient reverses. Consequently, this drives water stored in the fruit to flow “downwards” through the xylem to rehydrate the roots. This explanation is supported by analogous phenomena in other studies. For instance, a previous study [38] on *Avicennia marina* found that when hypersaline conditions limited root water uptake, nocturnal water absorption by the leaves drove a “top–down” rehydration of the plant via reverse sap flow.

This hydraulic redistribution has significant implications for both fruit quality and agricultural practices. By withdrawing water from the fruit, reverse sap flow can increase solute concentration, such as sugars, potentially enhancing fruit flavor and total soluble solids. This hypothesis is supported by research in grapes, where xylem backflow is a crucial pathway for water removal during ripening, a process directly linked to the accumulation of sugars and the overall quality of the berry [7]. Furthermore, our findings in watermelon provide a mechanistic explanation for the agricultural practice of withholding irrigation before harvest. For example, the technique of applying regulated deficit irrigation or ceasing irrigation entirely before harvest to enhance fruit sugar content is directly explained by our results. This practice induces a hydraulic deficit, in which the plant withdraws water from the fruit via reverse sap flow, thereby concentrating sugars and other quality-related solutes. Similar hydraulic behaviors have been inferred during the ripening of maize kernels and other fruits, suggesting that reverse sap flow from the fruit is a conserved and agriculturally significant mechanism.

While this study provides novel insights into reverse sap flow from the fruit, we acknowledge certain limitations that warrant further investigation. A primary limitation is the small sample size (n = 1 for each treatment) in the experiment. However, this hydraulic behaviour was consistently observed in a larger cohort of plants (n > 10) during our preliminary work, although it was not the focus of in-depth investigation at that time [37]. Therefore, this component could be interpreted as a compelling “proof-of-concept” case study that qualitatively elucidates the endogenous causes and exogenous triggers of reverse sap flow and demonstrates the fruit’s role as a water buffer, rather than as a statistically robust conclusion. Future studies employing larger replicates are necessary to statistically validate the observed phenomenon. Additionally, while conducting experiments under outdoor conditions enhances ecological relevance, it also introduces inherent environmental variability (e.g., microclimate, soil heterogeneity) that cannot be fully controlled. Finally, the scope of our study was to elucidate the hydraulic mechanisms; we did not quantify downstream impacts on fruit quality (e.g., total soluble solids, cracking incidence) or explore the underlying genetic architecture of this trait. These aspects represent critical and exciting directions for subsequent investigations.

In summary, our work systematically explains the causal mechanisms of reverse sap flow from the fruit and highlights its role in adaptive water redistribution under drought stress. These findings refine the source–sink theory in plant physiology by illustrating the dynamic transition of fruits between sink and source roles. Future studies should further explore the impact of reverse flow on fruit quality (e.g., sugar content) and investigate whether the trait of reverse flow is associated with specific plant genes. Such insights could provide valuable guidance for agricultural management and crop breeding strategies.

## 4. Materials and Methods

### 4.1. Plant-Wearable Sap Flow Sensor

The sap flow sensor used for measuring stem sap flow in plants was independently developed, as shown in Appendix A. This sensor consists of a three-layer structure: (1) a 10 µm thick stretchable polydimethylsiloxane (PDMS) base layer providing physical support while enabling robust and reversible adhesion to the plant surface; (2) a laser-patterned serpentine copper circuit (6 µm) laminated between two layers of polyimide (PI, ≈1 µm) forming the stretchable circuit for conductive communication; (3) a top sensing layer integrating two micro-temperature sensors (TMP117MAIDRVR) and a heater (FTP18BC330Q03RT) for sap flow measurement. The data measured by the sensor are wirelessly transmitted to a computer for data analysis and processing.

### 4.2. Plant Materials and Experimental Conditions

The experiment was conducted at Zhejiang University, Hangzhou, China (30.30° N, 120.08° E). Watermelon (cv. 8424) plants were cultivated outdoors in planter boxes (40 cm × 40 cm × 40 cm) filled with a commercial soil substrate (HAWITA) from April to August 2025. Healthy watermelon plants, 3 weeks post-fruit set, were selected for sensor installation at three key hydraulic positions: basal stem, leaf branch, and fruit branch. Sap flow sensors measured data at 10-min intervals, each lasting 8 min. Soil water content was monitored using soil moisture sensors (EC-5, METER, Pullman, WA, USA) uniformly installed at a depth of 10 cm and a horizontal distance of approximately 5 cm from the plant’s primary root system, while light intensity (SQ-521, METER, Pullman, WA, USA) and temperature/humidity (MX1101, Oneset, Cape Cod, MA, USA) sensors were placed above the plant to monitor environmental parameters synchronously during plant growth.

### 4.3. Drought Stress Contrast Experiment

Two healthy watermelon plants, growing consistently in a large planter box (40 cm × 40 cm × 80 cm), were selected for a drought stress experiment. The plants were carefully matched for initial size and vigour, both exhibited a stem diameter of approximately 5 mm and were confirmed to be free of any visible signs of disease or pest damage. To synchronize their reproductive development, female flowers at corresponding nodal positions on both plants were hand-pollinated on the same day. Three weeks post-fruit set, when the fruits had reached a uniform diameter of approximately 12 cm, the treatments were applied: one plant retained its fruit (fruit-bearing), while the fruit was removed from the other (defruited). A rainout shelter constructed from highly transparent waterproof film was used to prevent precipitation on the two plants. Sap flow sensors were installed at the key hydraulic positions (basal stem, leaf branch, fruit branch) of both plants. Irrigation was terminated after sensor deployment, with daily growth status recorded alongside continuous monitoring of sap flow and environmental parameters. The growth conditions of the two plants during the drought treatment are shown in Appendix A. Before the drought treatment, both plants had normal leaves (Day 1). On Day 6, most leaves near the apex of the plant without fruit were wilted, while the plant with fruit showed only slight leaf desiccation. By Day 8, the plant without fruit exhibited complete leaf desiccation, whereas the plant with fruit retained live leaves from the roots to the fruit region. On Day 12, the plant with fruit was completely dead.

## Figures and Tables

**Figure 1 plants-15-00105-f001:**
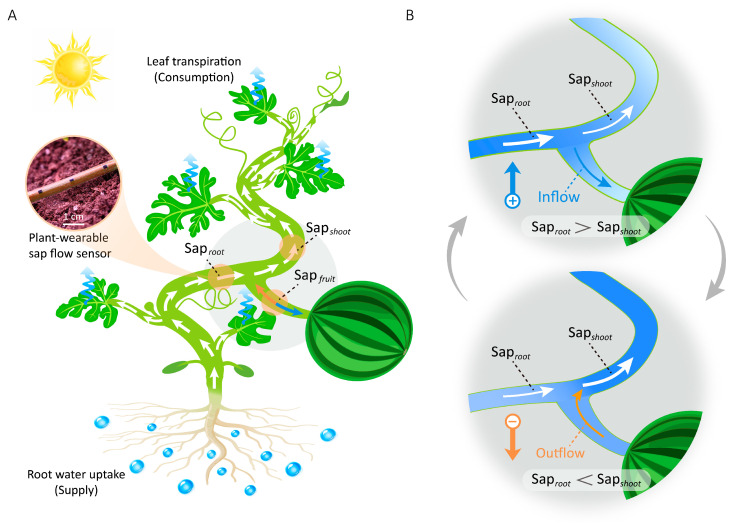
Relationship between plant water supply–consumption balance and sap flow direction of fruits. (**A**) Diagram illustrating the internal water transport pathways in watermelon plants. (**B**) Bidirectional fruit sap flow is regulated by water supply–consumption imbalance (Sap*_root_* vs. Sap*_shoot_*). Arrows within the plant indicate the direction of sap flow.

**Figure 2 plants-15-00105-f002:**
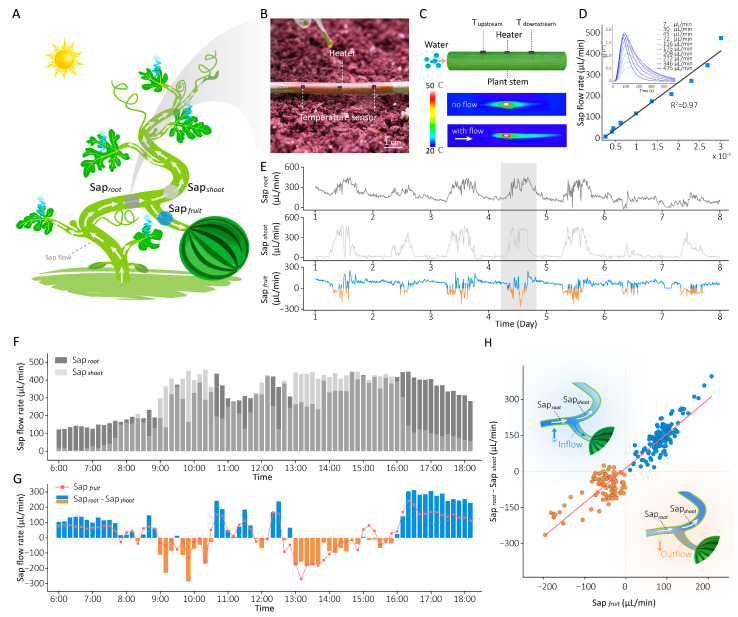
In situ monitoring of water distribution within watermelon plants. (**A**) Diagram showing the deployment of sap flow sensors at key points (basal stem, leaf branch, and fruit branch) on the watermelon plant. (**B**) Photograph of plant-wearable sensors attached to the stem of a watermelon plant. (**C**) Schematic diagram of the plant sap flow measurement principle. The infrared thermography images show the spatial temperature distribution in the absence (top) and presence (bottom) of flow when the heater is operating. (**D**) Linear regression between sap flow rates and differential temperature (ΔT) slope (the slope of the ΔT curve). The inset figure shows the ΔT between T_downstream_ and T_upstream_ in response to varying sap flow rates, with the curves smoothed using a moving average to reduce high-frequency noise and enhance visual clarity. (**E**) Long-term monitoring of stem sap flow rates at the three key points. Fluctuations during a typical daytime period (6:00–18:00) of (**F**) water supply rate (Sap*_root_*) and water consumption rate (Sap*_shoot_*), (**G**) fruit sap flow rate (red curve) and ΔSap (Sap*_root_* − Sap*_shoot_*, blue and orange bars). (**H**) The linear relationship between fruit sap flow rate and ΔSap.

**Figure 3 plants-15-00105-f003:**
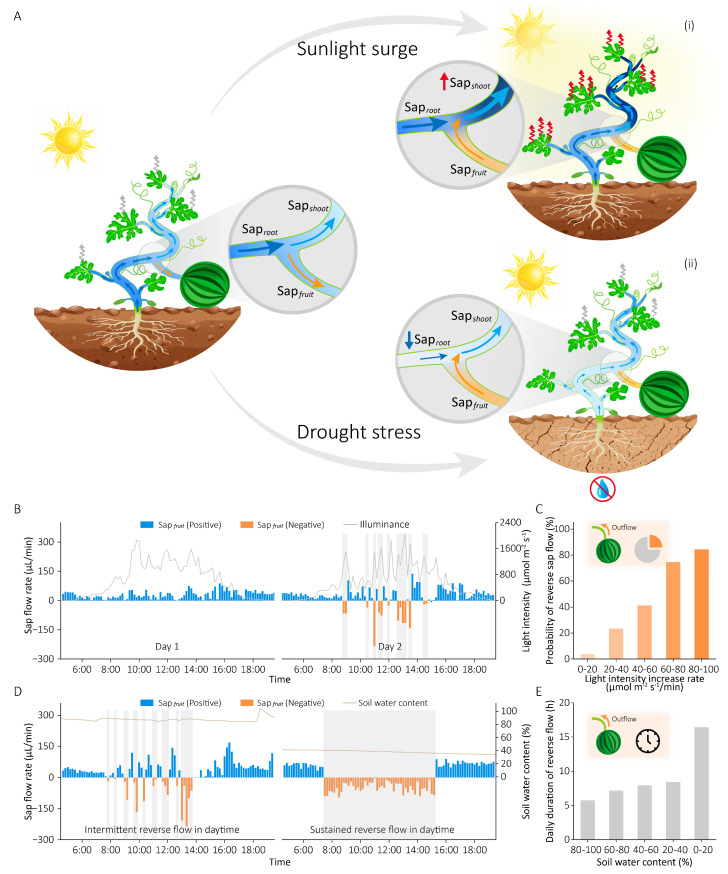
Environmental triggers of reverse sap flow in field-grown plants. (**A**) Water distribution at plants’ three positions under different scenarios: normal condition (left), during a light intensity surge (**i**), and under soil drought (**ii**). Wavy arrows denote leaf transpiration. (**B**) Fruit sap flow rate fluctuations respond to daytime light changes under adequate soil moisture conditions. (**C**) The relationship between the probability of reverse sap flow occurrence and the rate of increase in light intensity, derived from 25-day continuous measurements. (**D**) Comparative daytime fruit sap flow rates under hydrated versus drought conditions. (**E**) The relationship between the daily duration of reverse sap flow and soil water content.

**Figure 4 plants-15-00105-f004:**
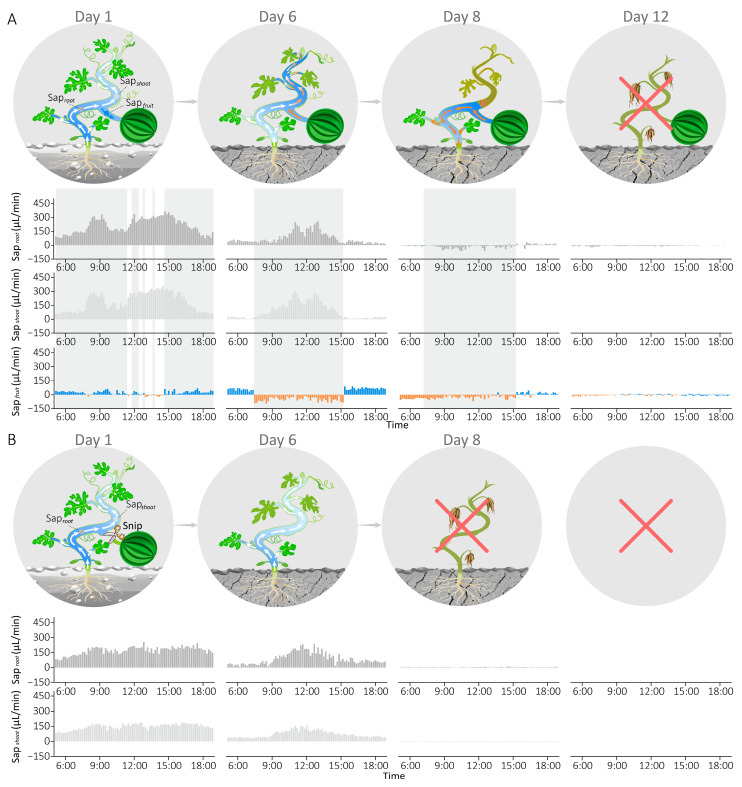
The impact of reverse sap flow from the fruit on plant drought resistance. (**A**) Growth status diagram and sap flow dynamics of the fruit-bearing plant during progressive drought: on Day 1, intermittent sap flow was observed during the daytime; on Day 6, sustained reverse sap flow occurred during high-light periods; on Day 8, Sap*_shoot_* became zero, and the vine between the fruit and apical shoot withered; on Day 12, the entire plant died. (**B**) Growth status diagram and sap flow dynamics of the fruit-removed plant under long-term drought stress: from Day 1 to Day 6, sap flow rates at both sites gradually decreased while maintaining Sap*_root_* > Sap*_shoot_* throughout; on Day 8, the plant died.

## Data Availability

The original contributions presented in this study are included in the article/Appendix A. Further inquiries can be directed to the corresponding author.

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
