# Peer review of "Fruit-Mediated Reverse Sap Flow: A Plant Water Balance Mechanism Enhancing Water Stress Adaptation"

_plants, 2025, doi:10.3390/plants15010105_

Round 1

Reviewer 1 Report

Comments and Suggestions for Authors

Overall Assessment

Your work provides a compelling mechanistic explanation of fruit-mediated reverse sap flow using plant-wearable sensors and synchronised environmental monitoring. The integration of continuous sap flow data with light intensity and soil moisture is a major strength. The drought comparison between fruit-bearing and defruited plants is particularly interesting and raises important physiological questions.

Pages 1–2:

The introduction effectively sets the scene by outlining the basics of sap flow and highlighting current technological limitations. However, it could be improved with a clearer explanation of the specific knowledge gap regarding the mechanisms behind reverse sap flow. Including a brief summary of what previous studies—such as those involving grapes, plums, and maize—have found would also be helpful, along with an explanation of why these findings have not yet addressed the fundamental mechanistic questions. Lastly, the introduction would be strengthened by a more explicit statement of your research objectives or hypotheses, ideally placed at the end of the section to better guide the reader’s expectations for the rest of the manuscript. 

  1. Methods

The methodology is generally well-structured, but several important clarifications are needed for reproducibility.

Sensor construction and calibration (p.11)

  • Please describe the calibration process in more detail.
  • Include accuracy, sensitivity, detection limit, and potential error sources.
  • Clarify whether ΔT measurements were corrected for environmental temperature fluctuations.

Experimental conditions

  • As plants were monitored outdoors, please describe how uncontrolled variations (e.g., cloudy days, rainfall) were accounted for.
  • The depth and consistency of soil moisture sensor placement across plants should be specified.

Drought stress contrast experiment (p.12)

  • Clarify the number of biological replicates used for both fruit-bearing and defruited treatments.
  • Provide information on fruit size or developmental stage to ensure comparability between plants.
  • Indicate whether stem diameter or overall plant vigour was matched before treatment.

Data processing and statistics

  • Please explain how the ΔSap metric (Sap_root – Sap_shoot) was calculated (time window, smoothing, or averaging).
  • Describe statistical tests used to validate correlations (e.g., Fig. 2H, R² = 0.87).
  • Indicate whether data were filtered or smoothed to reduce noise.

2.Results

Reverse sap flow dynamics (pp.4–5)

Your results clearly show the relationship between ΔSap and sap flow direction. To further enhance clarity:

  • Consider defining ΔSap earlier in the manuscript.
  • In Fig. 2E–G, contrast between colours could be improved, or add inset zoom-ins during periods of reverse flow.

Light-triggered effects (pp.6–7)

  • Please clarify how the “probability of reverse sap flow” was calculated.
  • Indicate how many days or datasets contributed to Fig. 3C.
  • It would be useful to define the threshold at which a light increase becomes “rapid” (e.g., >80 μmol m⁻² s⁻¹/min).

Drought-induced responses (pp.8–10)

  • The observation that sap flows from fruit to root under severe drought (Day 8, Fig. 4A) is intriguing. Consider elaborating on potential anatomical or physiological mechanisms.
  • As the survival difference between fruit-bearing and defruited plants is central to your argument, additional justification or discussion of variability would be helpful.
  1. Discussion

Interpretation

The discussion is cohesive, but a few areas could be expanded:

  1. Fruit physiology – briefly discuss whether reverse flow might influence fruit sugar content or internal water dynamics.
  2. Comparison with other crops – a short paragraph linking your findings to known reverse flow behaviour in grapes, plums, or maize would contextualise your results.
  3. Mechanism under extreme drought – consider a more detailed explanation of how water is redistributed to the roots.

Limitations

A short paragraph acknowledging limitations (e.g., outdoor variability, small sample size in the drought contrast) would enhance transparency and strengthen the manuscript.

Some questions that needs explanation

  1. Were there any instances of sensor drift or data drop-out during the monitoring period?
  2. Could phloem backflow contribute to the observed reverse transport under drought?
  3. Do you expect similar fruit-to-shoot or fruit-to-root redistribution patterns in other cucurbits?
  4. How scalable is the wearable sensor system for broader agricultural applications?

Comments on the Quality of English Language

The manuscript would benefit from moderate language polishing to improve clarity.

Author Response

Response to reviewers’ comments

Comments: Your work provides a compelling mechanistic explanation of fruit-mediated reverse sap flow using plant-wearable sensors and synchronised environmental monitoring. The integration of continuous sap flow data with light intensity and soil moisture is a major strength. The drought comparison between fruit-bearing and defruited plants is particularly interesting and raises important physiological questions.

Response: Thank you for reviewing our paper and your positive comments. We have carefully revised the manuscript based on your suggestions. We greatly appreciate your help in improving this manuscript.

  1. Pages 1–2:

The introduction effectively sets the scene by outlining the basics of sap flow and highlighting current technological limitations. However, it could be improved with a clearer explanation of the specific knowledge gap regarding the mechanisms behind reverse sap flow. Including a brief summary of what previous studies—such as those involving grapes, plums, and maize—have found would also be helpful, along with an explanation of why these findings have not yet addressed the fundamental mechanistic questions. Lastly, the introduction would be strengthened by a more explicit statement of your research objectives or hypotheses, ideally placed at the end of the section to better guide the reader’s expectations for the rest of the manuscript.

Response: Thanks for your advice, we have added a summary of observations in plums, grapes, and maize, and explained why these studies could not address the fundamental mechanistic questions. Besides, we have added the research objectives of this work in the last paragraph of the introduction.

Following corrections have been made in the manuscript

Add Text:For instance, Knoche et al. observed that xylem backflow from fruit to tree frequently occurred in the morning in developing European plums [6]. Keller et al. injected dye into grape berries, and documented the movement of xylem-mobile dye from the berries back to the shoot during the ripening stage for sugar accumulation [7]. Similarly, Zhang et al. employed a comparable method and found that a portion of water flowed back from the maize cob and ear to the plant via the xylem during the grain dehydration phase [8]. Although these studies observed reverse sap flow across multiple crop species, they were unable to explain the underlying causes of this phenomenon due to the inability to non-destructively, real-time, and accurately quantify the movement of sap flow in plants.(Introduction Section, Page 2, Line 45-54).

Add Text:Therefore, the objectives of this study were to elucidate the underlying causal mechanisms of reverse sap flow and to investigate its role in plant water physiology.(Introduction Section, Page 3, Line 93-94).

Delete:Despite these corroborating observations across multiple crop species, the mechanistic basis of this phenomenon remains unclear.(Introduction Section, Page 2, Line 54).

  1. Methods

The methodology is generally well-structured, but several important clarifications are needed for reproducibility.

(1) Sensor construction and calibration (p.11)

Please describe the calibration process in more detail.

Include accuracy, sensitivity, detection limit, and potential error sources.

Response: We have expanded the description of the sensor calibration in the Supplementary Materials (Figure S2) as suggested.

Following corrections have been made in the manuscript

Add Text:The syringe’s injection rate was controlled using a micro-injection pump (LSP01-1B, China), and different water injection rates were set to simulate sap flow through the stem. A sap flow sensor was installed at the middle section of the stem for measurement. The calibration procedure was as follows: First, the injection rate of the micro-injection pump was set from low to high, and the actual sap flow rate was calculated by measuring the total mass of water collected by a balance within a unit time. Second, at each sap flow rate, the sensor was used to repeatedly measure multiple sets of time-temperature difference (ΔT = Tdownstream − Tupstream) curves, from which the ascending slope of each curve was derived, and the mean and standard deviation were obtained. Third, the sap flow rates and the corresponding mean ascending slopes of the ΔT curves were plotted as scatter points (Figure 2E), and a linear regression was performed to obtain the calibration curve and the sap flow calculation equation. Finally, by comparing the sensor-predicted values with the known flow rates, the mean relative error (MRE) of the sensor measurements was 0.12, and the limit of detection (LOD) was calculated as 5 μL/min based on the 3-sigma rule. The primary source of measurement error in this sensor arises from whether the three components—the two temperature sensors and the heater—maintain tight contact with the stem surface during installation.(Supplementary Materials, Page 2, Figure S2).

Clarify whether ΔT measurements were corrected for environmental temperature fluctuations.

Response: Additional experiments were conducted to investigate the effect of environmental temperature on sap flow measurements. The sensor was used to measure at a constant flow rate (150 μL/min) under a range of temperatures from 15 ℃ to 50 ℃, which represents a typical thermal range in field conditions. As shown in Figure R1, the measured flow rates did not show a significant difference (ANOVA, N=32, P=0.079). These results indicate that the sensor’s measurements are robust against ambient temperature fluctuations.

Figure R1 Responses of the flow sensor under different room temperatures.

(2) Experimental conditions

As plants were monitored outdoors, please describe how uncontrolled variations (e.g., cloudy days, rainfall) were accounted for.

Response: All experiments were performed under outdoor conditions. The only exception was the drought stress experiment, during which a rainout shelter constructed from highly transparent waterproof film was used to prevent precipitation on the designated plants. We will provide additional details in the Materials and Methods section.

Following corrections have been made in the manuscript

Add Text:A rainout shelter constructed from highly transparent waterproof film was used to prevent precipitation on the two plants.(Materials and Methods Section, Page 12, Line 391-392).

The depth and consistency of soil moisture sensor placement across plants should be specified.

Response: To ensure measurement consistency, all soil moisture sensors were uniformly installed at a depth of 10 cm and a horizontal distance of approximately 5 cm from the plant’s primary root system. We will provide additional details in the Materials and Methods section.

Following corrections have been made in the manuscript

Revise Text:Soil water content was monitored using soil moisture sensors (EC-5, METER, USA) uniformly installed at a depth of 10 cm and a horizontal distance of approximately 5 cm from the plant’s primary root system.(Materials and Methods Section, Page 12, Line 376-379).

(3) Drought stress contrast experiment (p.12)

Clarify the number of biological replicates used for both fruit-bearing and defruited treatments.

Response: It should be acknowledged that a limitation of the current study is the small sample size for the drought stress comparison, which was conducted on only a single fruit-bearing plant and a single defruited plant. While we recognise that this small sample size (n=1 per treatment) limits the generalizability of the result, the finding that fruit presence enhances drought resistance and prolongs survival time is strongly supported by our preliminary observations from previous experiments. This aspect will be more rigorously investigated with a larger sample size in future work. We will discuss the limitations of the small sample size in the Discussion section.

Following corrections have been made in the manuscript

Add Text:While this study provides novel insights into fruit-mediated reverse sap flow, we acknowledge certain limitations that warrant further investigation. A primary limitation is the small sample size (n=1 for each treatment) in the experiment. However, this hydraulic behaviour was consistently observed in a larger cohort of plants (n > 10) during our preliminary work, although it was not the focus of in-depth investigation at that time [37]. Therefore, this component could be interpreted as a compelling “proof-of-concept” case study that qualitatively elucidates the endogenous causes and exogenous triggers of reverse sap flow and demonstrates the fruit’s role as a water buffer, rather than as a statistically robust conclusion. Future studies employing larger replicates are necessary to statistically validate the observed phenomenon.(Discussion Section, Page 11, Line 335-344).

Provide information on fruit size or developmental stage to ensure comparability between plants.

Response: The plants selected for the drought stress experiment were uniform in their developmental stage. Specifically, all plants were three weeks post-fruit set, with a fruit diameter of approximately 12 cm. This information will be added to the Materials and Methods section.

Following corrections have been made in the manuscript

Revise Text:To synchronize their reproductive development, female flowers at corresponding nodal positions on both plants were hand-pollinated on the same day. Three weeks post-fruit set, when the fruits had reached a uniform diameter of approximately 12 cm, the treatments were applied: one plant retained its fruit (fruit-bearing), while the fruit was removed from the other (defruited).(Materials and Methods Section, Page 12, Line 386-392)

Indicate whether stem diameter or overall plant vigour was matched before treatment.

Response: The plants used in the drought stress experiment were carefully matched for initial vigour. They all had a consistent stem diameter of approximately 5 mm and were confirmed to be healthy and free of any visible signs of disease or pest damage before the experiment. This will be clarified in the Materials and Methods.

Following corrections have been made in the manuscript

Add Text:The plants were carefully matched for initial size and vigour, both exhibited a stem diameter of approximately 5 mm and were confirmed to be free of any visible signs of disease or pest damage.(Materials and Methods Section, Page 12, Line 384-386)

(4) Data processing and statistics

Please explain how the ΔSap metric (Sap_root – Sap_shoot) was calculated (time window, smoothing, or averaging).

Response: The metric ΔSap was calculated as the instantaneous difference between the sap flow rate measured at the basal stem and that measured at the leaf branch. No additional smoothing or averaging was applied to the data for this calculation.

Describe statistical tests used to validate correlations (e.g., Fig. 2H, R² = 0.87).

Response: The correlation between ΔSap and Sap fruit was analyzed using a linear regression model. The coefficient of determination (R²) was calculated to be 0.87.

Indicate whether data were filtered or smoothed to reduce noise.

Response: Data smoothing was selectively applied. Specifically, the time-temperature difference curves presented in the inset of Figure 2D were smoothed using a moving average to reduce high-frequency noise and enhance visual clarity. All other data presented in the manuscript were not smoothed or filtered. This will be explicitly stated in the legend for Figure 2D.

Following corrections have been made in the manuscript

Add Text:with the curves smoothed using a moving average to reduce high-frequency noise and enhance visual clarity.(Results Section, Page 4, Line 134-135).

  1. Results

(1) Reverse sap flow dynamics (pp.4–5)

Your results clearly show the relationship between ΔSap and sap flow direction. To further enhance clarity:

Consider defining ΔSap earlier in the manuscript.

Response: Thank you for your valuable advice. We agree that defining ΔSap earlier enhances the clarity of the manuscript.

Following corrections have been made in the manuscript

Add Text:This establishes the supply-consumption difference, ΔSap = Sap root – Sap shoot, as a key indicator, where its sign dictates the direction of sap flow relative to the fruit.(Introduction Section, Page 3, Line 102-103).

In Fig. 2E–G, contrast between colours could be improved, or add inset zoom-ins during periods of reverse flow.

Response: Thank you for this constructive suggestion. To improve visual clarity, we have revised Figure 2E–G to provide better contrast between the datasets.

Following corrections have been made in the manuscript

Revise Figure 2EG:

Figure 2

(2) Light-triggered effects (pp.67)

Please clarify how the “probability of reverse sap flow” was calculated.

Response: We appreciate the opportunity to clarify this calculation. The probability of reverse sap flow was determined as follows:

  1. Light intensity was recorded at 10-minute intervals throughout the experiment.
  2. The rate of light intensity increase was calculated for each interval by taking the difference between the current measurement and the measurement from 10 minutes prior.
  3. These rates of increase were then categorized into predefined bins (e.g., 0-20, 20-40 μmol m⁻² s⁻¹/min).
  4. For each bin, we counted the total number of sap flow measurements (Ntotal) and the number of occurrences that were concurrent with an observed reverse sap flow event (Nreverse).
  5. The probability for each bin was then calculated as the ratio: Probability = Nreverse / Ntotal.

Indicate how many days or datasets contributed to Fig. 3C.

Response: The data presented in Figure 3C were compiled from a continuous measurement period of 25 days. We will add this information to the figure legend for clarity.

Following corrections have been made in the manuscript

Add Text:derived from 25-day continuous measurements.(Results Section, Page 7, Line 189-190).

It would be useful to define the threshold at which a light increase becomes “rapid” (e.g., >80 μmol m² s¹/min).

Response: Thank you for your advice. Following your suggestion, we have now explicitly defined a threshold for a “rapid” light intensity increase. Based on the analysis in Figure 3C, we designated 80 μmol m⁻² s⁻¹/min as the critical threshold. This value was chosen because when the rate of increase surpasses this point, the probability of inducing reverse sap flow rises dramatically to over 80%.

Following corrections have been made in the manuscript

Revise Text:As k reaches a critical threshold (= 80 μmol m⁻² s⁻¹/min), the subsequent rapid stomatal opening triggers sudden transpiration surges that markedly surpass root water supply capacity, resulting in a high probability (over 80%) of reverse sap flow from fruits.(Results Section, Page 7, Line 209-212).

(3) Drought-induced responses (pp.810)

The observation that sap flows from fruit to root under severe drought (Day 8, Fig. 4A) is intriguing. Consider elaborating on potential anatomical or physiological mechanisms.

Response: The fundamental driving force for water movement within a plant is the water potential gradient. Water invariably moves from areas of higher water potential to areas of lower water potential. Under severe drought, the plant root is unable to absorb water from the soil, causing the root water potential to drop to extremely low levels. When the root water potential falls substantially below that of the fruit, which acts as a storage capacitor, the water potential gradient between the fruit and the root reverses. Consequently, water stored in the fruit flows “downwards” through the xylem to rehydrate the roots. This explanation is supported by analogous phenomena in other studies. For instance, a previous study [R1] on Avicennia marina found that when hypersaline conditions limited root water uptake, leaves absorb water directly from the high-humidity environment during the night, causing an increase in leaf water potential, which in turn drove a top-down rehydration via reverse sap flow. This provides strong evidence that stored water in one organ can be redistributed to another, more water-stressed organ, supporting our interpretation. We have added additional elaboration in the Discussion section.

Reference: [R1] Coopman, R.E.; Nguyen, H.T.; Mencuccini, M.; Oliveira, R.S.; Sack, L.; Lovelock, C.E. and Ball, M.C. Harvesting water from unsaturated atmospheres: deliquescence of salt secreted onto leaf surfaces drives reverse sap flow in a dominant arid climate mangrove, Avicennia marina. New Phytol. 2021, 231, 1401-1414. https://doi.org/10.1111/nph.17461

Following corrections have been made in the manuscript

Add Text:The physiological basis for this observation lies in the fundamental principle of the water potential gradient, which dictates that water invariably moves from areas of higher to lower potential. Under severe drought, when the roots are unable to absorb water from the soil, their water potential plummets. Once the root water potential falls substantially below that of the fruit—which acts as a storage capacitor—the hydraulic gradient reverses. Consequently, this drives water stored in the fruit to flow “downwards” through the xylem to rehydrate the roots. This explanation is supported by analogous phenomena in other studies. For instance, a previous study [38] on Avicennia marina found that when hypersaline conditions limited root water uptake, nocturnal water absorption by the leaves drove a “top-down” rehydration of the plant via reverse sap flow.(Discussion Section, Page 10, Line 311-320).

Add Reference: the above-mentioned R1 is added as Ref. 38.

As the survival difference between fruit-bearing and defruited plants is central to your argument, additional justification or discussion of variability would be helpful.

Response: Thank you for this insightful comment. We agree that, given the small sample size (n=1 for each treatment), the conclusion regarding the survival difference warrants additional justification and a discussion of variability. We have addressed this in the revised Discussion section. We would first like to emphasize that this experiment was designed as a “proof-of-concept” case study, for which the two plants were meticulously matched (for size, health, developmental stage, and growing environment) to minimize confounding variability. The observed survival advantage of the fruit-bearing plant, while based on n=1, is strongly supported by three lines of evidence: (1) it corroborates our own repeated, preliminary observations from prior experiments; (2) it is consistent with the fundamental physiological principle of water-potential-driven flow; and (3) it aligns with existing literature on the role of fruits as water capacitors [R2]. We now explicitly acknowledge this limitation in the Discussion and state that future studies with larger replicates are needed to statistically validate this important phenomenon.

Reference: [R2] Shivashankar, S.; Sumathi, M.; Ugalat, J. Drought-stress-induced corky tissue formation in sapota fruit cv. Cricket ball is linked to enhanced “reverse flow”. Sci. Hortic. 2014, 169, 20-26. https://doi.org/10.1016/j.scienta.2014.01.047.

Following corrections have been made in the manuscript

Add Text:While this study provides novel insights into fruit-mediated reverse sap flow, we acknowledge certain limitations that warrant further investigation. A primary limitation is the small sample size (n=1 for each treatment) in the experiment. However, this hydraulic behaviour was consistently observed in a larger cohort of plants (n > 10) during our preliminary work, although it was not the focus of in-depth investigation at that time [37]. Therefore, this component could be interpreted as a compelling “proof-of-concept” case study that qualitatively elucidates the endogenous causes and exogenous triggers of reverse sap flow and demonstrates the fruit’s role as a water buffer, rather than as a statistically robust conclusion. Future studies employing larger replicates are necessary to statistically validate the observed phenomenon.(Discussion Section, Page 11, Line 335-344).

  1. Discussion

Interpretation

The discussion is cohesive, but a few areas could be expanded:

(1)  Fruit physiology briefly discuss whether reverse flow might influence fruit sugar content or internal water dynamics.

Response: Thank you for this excellent suggestion. We agree that exploring the implications for fruit quality significantly strengthens the discussion. We have now incorporated a new paragraph discussing the impact of reverse sap flow on fruit sugar content.

Following corrections have been made in the manuscript

Revise Text:This hydraulic redistribution has significant implications for both fruit quality and agricultural practices. By withdrawing water from the fruit, reverse sap flow can increase the concentration of solutes, such as sugars, potentially enhancing fruit flavor and total soluble solids. This hypothesis is supported by research in grapes, where xylem backflow is a crucial pathway for water removal during ripening, a process directly linked to the accumulation of sugars and the overall quality of the berry [7].(Discussion Section, Page 11, Line 321-326).

(2)  Comparison with other crops a short paragraph linking your findings to known reverse flow behaviour in grapes, plums, or maize would contextualise your results.

Response: We appreciate this suggestion to link our findings to known reverse flow behaviors in other crops. A new paragraph has been added to the Discussion.

Following corrections have been made in the manuscript

Revise Text:This hydraulic redistribution has significant implications for both fruit quality and agricultural practices. By withdrawing water from the fruit, reverse sap flow can increase the concentration of solutes, such as sugars, potentially enhancing fruit flavor and total soluble solids. This hypothesis is supported by research in grapes, where xylem backflow is a crucial pathway for water removal during ripening, a process directly linked to the accumulation of sugars and the overall quality of the berry [7]. Furthermore, our findings in watermelon provide a mechanistic explanation for the agricultural practice of withholding irrigation before harvest. For example, the technique of applying regulated deficit irrigation or ceasing irrigation entirely before harvest to enhance fruit sugar content is directly explained by our results. This practice induces a hydraulic deficit where the plant withdraws water from the fruit via reverse sap flow, thereby concentrating sugars and other quality-related solutes. Similar hydraulic behaviors have been inferred during the ripening of maize kernels and other fruits, suggesting that fruit-mediated reverse flow is a conserved and agriculturally significant mechanism.(Discussion Section, Page 11, Line 321-334).

(3)  Mechanism under extreme drought consider a more detailed explanation of how water is redistributed to the roots.

Response: We believe this mechanism is detailed in the newly added text in the fourth paragraph of the Discussion section. In this section, we explain that under severe drought, the root water potential plummets, reversing the hydraulic gradient between the fruit (a high-potential source) and the root (a low-potential sink). This reversed gradient is what drives water “downwards” through the xylem to rehydrate the roots. We have double-checked this paragraph for clarity and believe it comprehensively covers the mechanism as requested.

Following corrections have been made in the manuscript

Add Text:The physiological basis for this observation lies in the fundamental principle of the water potential gradient, which dictates that water invariably moves from areas of higher to lower potential. Under severe drought, when the roots are unable to absorb water from the soil, their water potential plummets. Once the root water potential falls substantially below that of the fruit—which acts as a storage capacitor—the hydraulic gradient reverses. Consequently, this drives water stored in the fruit to flow “downwards” through the xylem to rehydrate the roots. This explanation is supported by analogous phenomena in other studies. For instance, a previous study [38] on Avicennia marina found that when hypersaline conditions limited root water uptake, nocturnal water absorption by the leaves drove a “top-down” rehydration of the plant via reverse sap flow.(Discussion Section, Page 10, Line 311-320).

  1. Limitations

A short paragraph acknowledging limitations (e.g., outdoor variability, small sample size in the drought contrast) would enhance transparency and strengthen the manuscript.

Response: Thank you for this essential advice. We fully agree that acknowledging the study’s limitations enhances its transparency and scientific rigor. We have now added a paragraph regarding the limitations in the Discussion.

Following corrections have been made in the manuscript

Add Text:While this study provides novel insights into fruit-mediated reverse sap flow, we acknowledge certain limitations that warrant further investigation. A primary limitation is the small sample size (n=1 for each treatment) in the experiment. However, this hydraulic behaviour was consistently observed in a larger cohort of plants (n > 10) during our preliminary work, although it was not the focus of in-depth investigation at that time [37]. Therefore, this component could be interpreted as a compelling “proof-of-concept” case study that qualitatively elucidates the endogenous causes and exogenous triggers of reverse sap flow and demonstrates the fruit’s role as a water buffer, rather than as a statistically robust conclusion. Future studies employing larger replicates are necessary to statistically validate the observed phenomenon. Additionally, while conducting experiments under outdoor conditions enhances ecological relevance, it also introduces inherent environmental variability (e.g., microclimate, soil heterogeneity) that cannot be fully controlled. Finally, the scope of our study was focused on elucidating the hydraulic mechanisms; we did not quantify the downstream impacts on fruit quality (e.g., total soluble solids, cracking incidence) or explore the underlying genetic architecture of this trait. These aspects represent critical and exciting directions for subsequent investigations.(Discussion Section, Page 11, Line 335-350).

Some questions that need explanation

(1)  Were there any instances of sensor drift or data drop-out during the monitoring period?

Response: We can confirm that the sensor system demonstrated high stability and data continuity throughout the long-term monitoring period. There was only one instance of data drop-out, lasting approximately 5 hours during the 4-week monitoring period, which occurred due to a short circuit caused by rainwater. This brief data gap did not coincide with the key environmental trigger events analyzed in the manuscript and did not impact the overall interpretation of the observed phenomena or the conclusions drawn.

(2)  Could phloem backflow contribute to the observed reverse transport under drought?

Response: This is a very insightful question regarding the specific transport pathway. While we cannot entirely exclude any minor phloem contribution, the primary driver of the observed reverse flow is almost certainly the xylem. The established role of phloem transport to the fruit is not only to deliver sugars but also to hydraulically buffer and isolate the fruit from short-term water potential fluctuations in the parent plant, preventing damage to the fruit caused by plant dehydration under drought conditions. Therefore, a significant phloem backflow under drought would be unlikely, as it would exacerbate fruit dehydration rather than alleviate stress on the vegetative organs. This view is strongly supported by published research in grapes [R3]. Thus, we posit that the phenomenon we observed is a xylem-mediated emergency water redistribution, which is functionally distinct from the primary buffering role of the phloem.

Reference: [R3] Choat, B.; Gambetta, G.; Shackel, K.; Matthews, M. Vascular Function in Grape Berries across Development and Its Relevance to Apparent Hydraulic Isolation. Plant Physiol. 2009, 151, 1677-1687. https://doi.org/10.1104/pp.109.143172

(3)  Do you expect similar fruit-to-shoot or fruit-to-root redistribution patterns in other cucurbits?

Response: We hypothesize that similar redistribution patterns are indeed likely to occur in other cucurbits, particularly in species known for drought tolerance. The role of a large, fleshy fruit as a temporary water capacitor is likely a conserved adaptive mechanism rather than a unique feature of watermelon. This concept aligns with the broader principle of organ-to-organ hydraulic redistribution under stress. As we discussed with the example of Avicennia marina [R1], where leaves facilitated a “top-down” rehydration, plants have evolved diverse strategies to move water from a temporary source to a critical sink. Therefore, observing a similar fruit-to-plant redistribution in other cucurbits with analogous fruit morphology and physiology seems highly probable.

Reference: [R1] Coopman, R.E.; Nguyen, H.T.; Mencuccini, M.; Oliveira, R.S.; Sack, L.; Lovelock, C.E. and Ball, M.C. Harvesting water from unsaturated atmospheres: deliquescence of salt secreted onto leaf surfaces drives reverse sap flow in a dominant arid climate mangrove, Avicennia marina. New Phytol. 2021, 231, 1401-1414. https://doi.org/10.1111/nph.17461

(4)  How scalable is the wearable sensor system for broader agricultural applications?

Response: Thank you for inquiring about the future potential of our technology. The current iteration of our wearable sensor system was designed primarily as a high-resolution research tool to uncover novel physiological phenomena. However, we believe its potential for broader agricultural applications is significant and scalable, primarily in three areas:

  • Precision Irrigation: By providing direct, real-time plant water status instead of relying solely on soil moisture or weather data, these sensors could trigger irrigation with high accuracy, saving water and preventing drought stress.
  • High-Throughput Phenotyping: Large-scale deployment of this wearable sensor system could be implemented in breeding programs to screen plant genotypes for superior drought tolerance traits.
  • Plant Health Diagnosis: Plant water physiology is closely linked to overall plant health. Therefore, continuous monitoring of plant water status can serve as a valuable tool for assessing overall plant growth and vitality.

Reviewer 2 Report

Comments and Suggestions for Authors

The manuscript by Chai et al. is a nice piece of experimental plant physiology. The authors used their innovative plant-wearable sensor to monitor the rate and the direction of sap flow in plants in near real-time. As a result, they pinpointed novel phenomena that were never recorded before like back flow of sap from fruit to other parts of the plant. The nicely connected the observed patterns of sap flow with the modelled stress and diurnal changes obtaining conclusive results. Overall, this study serves as a convincing proof-of-concept for the developed technology of non-invasive sensing of sap flow. The only weakness of this study is the small number of replicas which could be amended by using several plants in parallel or by setting up sequential experiments if parallel measurements are not achievable at this stage. The authors should explicitly mention and discuss limitations stemming from these circumstances. Otherwise this paper is ripe for publication.

Author Response

Response to reviewers’ comments

Comments: The manuscript by Chai et al. is a nice piece of experimental plant physiology. The authors used their innovative plant-wearable sensor to monitor the rate and the direction of sap flow in plants in near real-time. As a result, they pinpointed novel phenomena that were never recorded before like back flow of sap from fruit to other parts of the plant. The nicely connected the observed patterns of sap flow with the modelled stress and diurnal changes obtaining conclusive results. Overall, this study serves as a convincing proof-of-concept for the developed technology of non-invasive sensing of sap flow. The only weakness of this study is the small number of replicas which could be amended by using several plants in parallel or by setting up sequential experiments if parallel measurements are not achievable at this stage. The authors should explicitly mention and discuss limitations stemming from these circumstances. Otherwise this paper is ripe for publication.

Response: We are very grateful for your positive evaluation of our work and for this constructive and crucial feedback. We fully agree that the small sample size (n=1 per treatment) limits the generalizability of the result.

While logistical and seasonal constraints prevent us from supplementing this with new replicates at present, we wish to emphasize that the core phenomenon is not an isolated finding. As we have cited in the manuscript, this hydraulic behaviour was consistently observed in a larger cohort of plants (n > 10) during our preliminary work [R1], although it was not the focus of in-depth investigation at that time. Furthermore, the finding is strongly supported by the fundamental physiological principle of water-potential-driven flow. Therefore, this component could be interpreted as a compelling “proof-of-concept” case study that qualitatively elucidates the endogenous causes and exogenous triggers of reverse sap flow and demonstrates the fruit’s role as a water buffer, rather than as a statistically robust conclusion.

In direct response to your suggestion, we have now added a paragraph to the Discussion, which explicitly addresses this point and states that future studies with larger replicates are necessary to statistically validate this important phenomenon. Thank you once again for your insightful guidance, which has significantly improved the transparency and rigor of our manuscript.

Reference: [R4] Zhang, R.; Chai, Y.; Liang, X.; Liu, X.; Wang, X.; Hu, Z. A New Plant-Wearable Sap Flow Sensor Reveals the Dynamic Water Distribution during Watermelon Fruit Development. Horticulturae 2024, 10. https://doi.org/10.3390/horticulturae10060649

Following corrections have been made in the manuscript

Add Text:While this study provides novel insights into fruit-mediated reverse sap flow, we acknowledge certain limitations that warrant further investigation. A primary limitation is the small sample size (n=1 for each treatment) in the experiment. However, this hydraulic behaviour was consistently observed in a larger cohort of plants (n > 10) during our preliminary work, although it was not the focus of in-depth investigation at that time [37]. Therefore, this component could be interpreted as a compelling “proof-of-concept” case study that qualitatively elucidates the endogenous causes and exogenous triggers of reverse sap flow and demonstrates the fruit’s role as a water buffer, rather than as a statistically robust conclusion. Future studies employing larger replicates are necessary to statistically validate the observed phenomenon. Additionally, while conducting experiments under outdoor conditions enhances ecological relevance, it also introduces inherent environmental variability (e.g., microclimate, soil heterogeneity) that cannot be fully controlled. Finally, the scope of our study was focused on elucidating the hydraulic mechanisms; we did not quantify the downstream impacts on fruit quality (e.g., total soluble solids, cracking incidence) or explore the underlying genetic architecture of this trait. These aspects represent critical and exciting directions for subsequent investigations.(Discussion Section, Page 11, Line 335-350).
